# A phase 1b randomised controlled trial of a glucagon-like peptide-1 and glucagon receptor dual agonist IBI362 (LY3305677) in Chinese patients with type 2 diabetes

Hongwei Jiang[1], Shuguang Pang[2], Yawei Zhang[3], Ting Yu[4], Meng Liu[4], Huan Deng[4], Li Li[4], Liqi Feng[4], Baili Song[4], Han Han-Zhang[4], Qingyang Ma[4], Lei Qian [4✉] & Wenying Yang [5✉]

The success of glucagon-like peptide-1 (GLP-1) receptor agonists to treat type 2 diabetes (T2D) and obesity has sparked considerable efforts to develop next-generation co-agonists that are more effective. We conducted a randomised, placebo-controlled phase 1b study (ClinicalTrials.gov: NCT04466904) to evaluate the safety and efficacy of IBI362 (LY3305677), a GLP-1 and glucagon receptor dual agonist, in Chinese patients with T2D. A total of 43 patients with T2D were enrolled in three cohorts in nine study centres in China and randomised in each cohort to receive once-weekly IBI362 (3.0 mg, 4.5 mg or 6.0 mg), placebo or open-label dulaglutide (1.5 mg) subcutaneously for 12 weeks. Forty-two patients received the study treatment and were included in the analysis, with eight receiving IBI362, four receiving placebo and two receiving dulaglutide in each cohort. The patients, investigators and study site personnel involved in treating and assessing patients in each cohort were masked to IBI362 and placebo allocation. Primary outcomes were safety and tolerability of IBI362. Secondary outcomes included the change in glycated haemoglobin $A_{1c}$ (HbA$_{1c}$), fasting plasma glucose (FPG) and post-mixed-meal tolerance test (post-MTT) glucose levels. IBI362 was well tolerated. Most commonly-reported treatment-emergent adverse events were diarrhoea (29.2% for IBI362, 33.3% for dulaglutide, 0% for placebo), decreased appetite (25.0% for IBI362, 16.7% for dulaglutide, 0% for placebo) and nausea (16.7% for IBI362, 16.7% for dulaglutide and 8.3% for placebo). HbA$_{1c}$, FPG and post-MTT glucose levels were reduced from baseline to week 12 in patients receiving IBI362 in all three cohorts. IBI362 showed a favourable safety profile and clinically meaningful reductions in blood glucose in Chinese patients with T2D.

[1] The First Affiliated Hospital and Clinical Medicine College, Henan University of Science and Technology, Luoyang, China. [2] Department of Endocrinology, Jinan Central Hospital, Jinan, China. [3] Department of Endocrinology, Pingxiang People's Hospital, Pingxiang, China. [4] Innovent Biologics, Inc, Suzhou, China. [5] Department of Endocrinology, China-Japan Friendship Hospital, Beijing, China. ✉email: cnradium@126.com; ywying_1010@126.com

Type 2 diabetes (T2D), characterized by hyperglycaemia, impairment of insulin secretion and insulin resistance, is a global health crisis with ever-growing incidence and prevalence. An estimated 463 million adults lived with diabetes in 2019, more than 90% of whom had T2D, with a projected increase to 700 million by 2045[1]. China has the largest number of adults with diabetes in 2019 and is estimated to top the list in the coming decades[1]. Often accompanied and exacerbated by obesity and overweight, T2D leads to a series of microvascular and macrovascular complications, which cause profound distress to patients and impose a huge burden on the healthcare system[2].

State-of-the-art T2D therapeutics, namely glucagon-like peptide-1 (GLP-1) receptor agonists and sodium-glucose cotransporter-2 (SGLT-2) inhibitors, are used either alone or as an add-on to standard-of-care treatment, achieve desirable glycaemic control targets, confer non-glycaemic benefits of body weight loss and blood pressure reduction, and reduce the risk of atherosclerotic cardiovascular disease, congestive heart failure and chronic kidney disease[3]. Despite the robustness of GLP-1 receptor mono-agonists, the change in gut hormone milieu accompanying substantial glycaemic improvements in patients undergoing bariatric surgery implied that a combination of gastrointestinal hormones may prove additive or even synergistic therapeutic outcomes[4–6]. Unimolecular poly-agonists with balanced activities at multiple gastrointestinal hormone receptors emerge as the promising next-generation therapeutics for the treatment of T2D and metabolic disorders[7].

Oxyntomodulin (OXM), a gut hormone that activates both the GLP-1 receptor and glucagon receptor[8], has been proved to increase energy expenditure while reducing energy intake. The anorectic and energy expenditure-promoting effects of OXM are mediated by activation of GLP-1 receptor and glucagon receptor, respectively[8,9]. When administered exogenously, OXM can improve glucose tolerance and result in body weight loss, making GLP-1 and glucagon receptor dual agonists a promising treatment option for patients with T2D and/or obesity. However, the inherited stimulation of gluconeogenesis and glycogenolysis by glucagon requires the optimal balancing of GLP-1 and glucagon receptor agonism[8]. Indeed, several dual agonists of this class have yet to demonstrate satisfactory efficacy on both glucose reduction and body weight loss, albeit with overall favourable safety profiles[10–12].

IBI362 (also known as LY3305677) is a once-weekly synthetic peptide analogue of mammalian OXM, with a fatty-acyl moiety to extend the half-life[13]. IBI362 potently bound to human and mouse GLP-1 receptors and glucagon receptors in vitro. In mice, IBI362 improved glucose control, decreased body weight in both *Gcgr* knockout (KO) and *Glp1r* KO settings and increased energy expenditure[14]. Furthermore, a first-in-human single-ascending-dose study in healthy subjects and a multiple-ascending-dose study in Chinese adults with overweight and obesity demonstrated the favourable safety profile, weight loss efficacy and multiple metabolic benefits of IBI362[13,15].

Here, in a randomised, placebo-controlled, multiple-ascending-dose phase 1b study, adopting the same dose regimen as the previous phase 1b study in Chinese participants with overweight or obesity[15], we evaluate the safety, tolerability, pharmacokinetics and efficacy of IBI362 in Chinese patients with T2D. IBI362 is well-tolerated, shows an overall favourable safety profile and demonstrates clinically meaningful glycemic control and weight loss, as well as multiple metabolic improvements in Chinese patients with T2D.

## Results

**Patients.** Between September 12th, 2020 and May 28th, 2021, 85 patients were screened for eligibility, of whom 43 were enrolled and randomised. Forty-two patients received study drug treatment, with eight receiving IBI362, four receiving placebo and two receiving dulaglutide in each cohort. Collectively, 38 patients (90.5%) completed the study and 37 (88.1%) completed treatment (Fig. 1). One patient missed the week 12 dosing due to the COVID-19 pandemic but attended the subsequent study visits. Patients receiving at least one dose of the study drug were

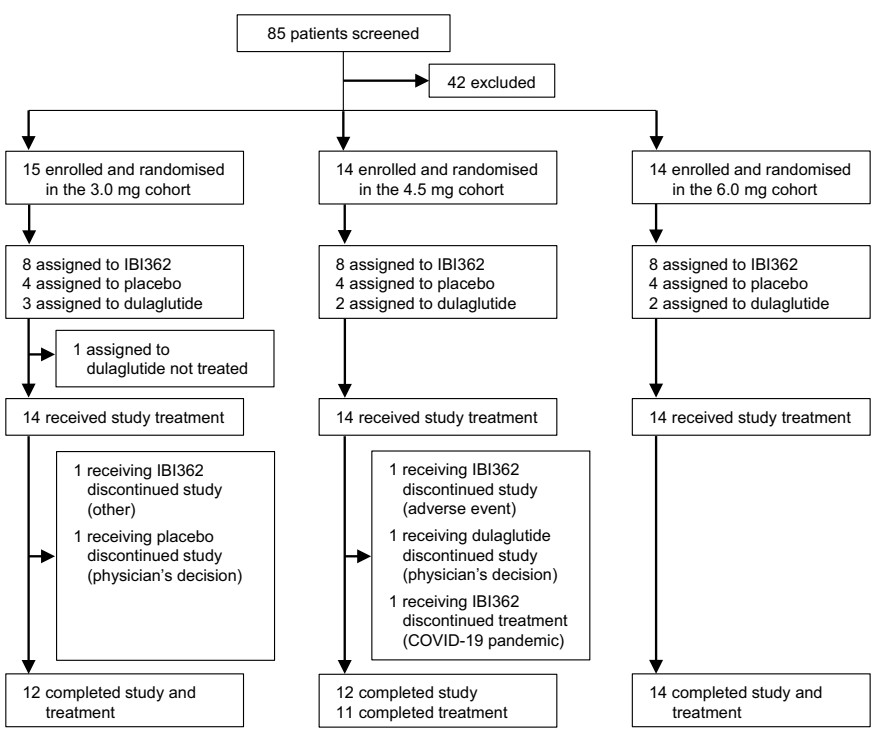

**Fig. 1 Patient flow.** The flow diagram shows the disposition of patients with type 2 diabetes screened and enroled.

included in the safety and efficacy population ($n = 42$). All patients receiving IBI362 were included in the pharmacokinetic population ($n = 24$). All patients receiving IBI362 or placebo were included in the immunogenicity population ($n = 36$). Demographic and baseline characteristics were essentially balanced between treatment groups (Table 1).

**Safety**. IBI362 was well tolerated and showed a good safety profile (Tables 2 and S3). One patient receiving IBI362 in the 4.5 mg cohort had mild decreased appetite and discontinued from the study. Serious TEAEs were reported in two patients receiving IBI362 in the 3.0 mg cohort, both unrelated to the study drug as judged by the investigators. All TEAEs were mild or moderate in severity.

Most commonly-reported TEAEs by SOC were gastrointestinal disorders (reported in 11 patients [45.8%] receiving IBI362, two [33.3%] receiving dulaglutide and two [16.7%] receiving placebo); by preferred term were diarrhoea (in seven patients [29.2%] receiving IBI362, two [33.3%] receiving dulaglutide and 0% receiving placebo), decreased appetite (in six patients [25.0%] receiving IBI362, one [16.7%] receiving dulaglutide and 0% receiving placebo) and nausea (in four patients [16.7%] receiving IBI362, one [16.7%] receiving dulaglutide and one [8.3%] receiving placebo) (Table 2 and S3). Gastrointestinal disorders and decreased appetite occurred more frequently in patients receiving IBI362 in the 4.5 mg and 6.0 mg cohorts. One patient receiving IBI362 had recurrent episodes of mild-to-moderate diarrhoea, continued the study without dose adjustment, and recovered after the last dose. Except for this, all other gastrointestinal disorders and decreased appetite reported in patients receiving IBI362 were transient and mild in severity. The median time to resolution of gastrointestinal symptoms (diarrhoea, nausea and vomiting) ranged from 1 to 4 days across IBI362 treatment groups and was 3 days for patients receiving dulaglutide.

Cardiac disorders were reported in five patients (20.8%) receiving IBI362, one (16.7%) receiving placebo and one (8.3%) receiving dulaglutide. Two patients receiving IBI362 in the 3.0 mg cohort reported serious adverse events of myocardial ischemia in the safety follow-up period, unrelated to the study treatment as judged by the investigators. One patient with serious adverse event of myocardial ischemia also reported non-serious adverse event of arrhythmia, specifically frequent premature ventricular

contractions, moderate in severity. One patient receiving IBI362 in the 4.5 mg cohort reported mild ventricular extrasystoles. Two patients receiving IBI362 in the 6.0 mg cohort reported mild atrioventricular block first degree (Table 2 and S3). Except for the serious adverse events, all cardiac disorders reported in patients receiving IBI362 were revealed by electrocardiogram, asymptomatic, and recovered within the study duration without dose adjustment and medical intervention.

Heart rate increase was evident in patients receiving IBI362 and dulaglutide. The magnitude of the heart rate increase was similar in patients receiving IBI362 in the 3.0 mg and 4.5 mg cohorts and those receiving dulaglutide (mean change from baseline of <10 beats per minute), and slightly higher in patients receiving IBI362 in the 6.0 mg cohort (mean change from baseline of <15 beats per minute) (Fig. S1).

Two patients receiving IBI362 in the 6.0 mg cohort and one receiving placebo reported adverse event of lipase increased. No investigator-suspected pancreatitis was reported. There were no reports of thyroid tumours, neoplasms or C-cell hyperplasia events.

Alanine aminotransferase and aspartate aminotransferase levels decreased gradually from baseline to week 12 (Fig. S2). One patient receiving IBI362 in the 3.0 mg cohort experienced a transient and asymptomatic increase in alanine aminotransferase level (within the 1.5-fold upper limit of normal), which was reported as an adverse event of hepatic function abnormal.

**Pharmacokinetics**. Table S1 showed non-compartmental pharmacokinetic parameters following subcutaneous administration of the first doses. IBI362 demonstrated slow absorption with peak concentrations ($C_{max}$) achieved ranging from 23.3 hours to 144.2 hours, and the median $T_{max}$ is ~72 hours. Once past the $T_{max}$, IBI362 concentrations declined slowly over several weeks with half-life ($t_{1/2}$) ranging from 147.3 hours to 673.3 hours (6.1 days to 28.1 days). Four patients (16.7%) receiving IBI362 and one (8.3%) receiving placebo developed anti-IBI362 antibody. No neutralizing antibodies were detected in post-baseline samples.

**Secondary and exploratory efficacy outcomes**. IBI362 treatment reduced $HbA_{1c}$ and FPG levels. Mean change from baseline to week 12 in $HbA_{1c}$ levels were $-1.46\%$ (SE 0.43), $-2.23\%$ (0.43) and $-1.66\%$ (0.41) for patients receiving IBI362 in the 3.0 mg,

**Table 1 Patients demographics and baseline characteristics.**

| | IBI362 3.0 mg ($n = 8$) | IBI362 4.5 mg ($n = 8$) | IBI362 6.0 mg ($n = 8$) | Dulaglutide ($n = 6$) | Pooled placebo ($n = 12$) |
|---|---|---|---|---|---|
| Sex | | | | | |
| Male | 6 (75.0%) | 3 (37.5%) | 5 (62.5%) | 6 (100%) | 6 (50.0%) |
| Female | 2 (25.0%) | 5 (62.5%) | 3 (37.5%) | 0 | 6 (50.0%) |
| Age (years) | 58.6 (5.3) | 47.9 (8.9) | 54.6 (10.6) | 50.7 (4.8) | 50.2 (8.5) |
| Race, Asian | 8 (100%) | 8 (100%) | 8 (100%) | 6 (100%) | 12 (100%) |
| $HbA_{1c}$ (%) | 8.9 (0.7) | 8.8 (1.0) | 8.5 (1.1) | 8.3 (1.4) | 8.3 (0.7) |
| Fasting plasma glucose (mmol/L) | 11.1 (1.9) | 10.7 (1.8) | 11.7 (2.2) | 11.2 (3.1) | 11.2 (2.6) |
| Body weight (kg) | 65.5 (9.2) | 70.9 (14.5) | 69.9 (9.7) | 75.1 (10.2) | 68.2 (11.5) |
| BMI (kg/m²) | 24.1 (1.4) | 25.7 (3.4) | 26.0 (2.2) | 26.6 (3.0) | 26.5 (2.7) |
| Blood pressure (mm Hg) | | | | | |
| Systolic | 129.9 (14.3) | 120.8 (11.0) | 122.1 (15.9) | 115.8 (10.1) | 121.3 (12.4) |
| Diastolic | 82.3 (12.2) | 81.9 (7.6) | 75.3 (9.0) | 76.3 (7.8) | 80.5 (7.8) |
| Heart rate (beats/min) | 66.0 (9.7) | 70.3 (11.2) | 65.9 (10.2) | 70.0 (11.8) | 71.1 (7.2) |
| Diabetes duration (years) | 2.1 (1.2-2.5) | 4.7 (2.2-5.5) | 6.0 (2.7-9.5) | 3.3 (1.6-3.4) | 3.1 (2.4-5.7) |
| Metformin use | 3 (37.5%) | 4 (50.0%) | 4 (50.0%) | 4 (66.7%) | 7 (58.3%) |

Data are presented as mean (SD), median (interquartile range) or n (%). BMI, body-mass index, $HbA_{1c}$, glycated haemoglobin $A_{1c}$.

**Table 2 Treatment-emergent adverse events.**

| | IBI362 3.0 mg (n = 8) | IBI362 4.5 mg (n = 8) | IBI362 6.0 mg (n = 8) | Dulaglutide (n = 6) | Pooled placebo (n = 12) |
|---|---|---|---|---|---|
| *Summary of treatment-emergent adverse events* | | | | | |
| Treatment-emergent adverse events | 6 (75.0) | 8 (100) | 7 (87.5) | 4 (66.7) | 9 (75.0) |
| Severe treatment-emergent adverse events | 0 | 0 | 0 | 0 | 0 |
| Serious treatment-emergent adverse events | 2 (25.0) | 0 | 0 | 0 | 0 |
| *Treatment-emergent adverse events of clinical interest** | | | | | |
| Gastrointestinal disorders | 2 (25.0) | 5 (62.5) | 4 (50.0) | 2 (33.3) | 2 (16.7) |
| Diarrhoea | 2 (25.0) | 3 (37.5) | 2 (25.0) | 2 (33.3) | 0 |
| Nausea | 1 (12.5) | 1 (12.5) | 2 (25.0) | 1 (16.7) | 1 (8.3) |
| Gastrooesophageal reflux disease | 0 | 2 (25.0) | 0 | 0 | 1 (8.3) |
| Abdominal distension | 0 | 2 (25.0) | 0 | 0 | 0 |
| Vomiting | 0 | 1 (12.5) | 0 | 1 (16.7) | 0 |
| Abdominal discomfort | 0 | 0 | 1 (12.5) | 0 | 0 |
| Dyspepsia | 0 | 0 | 0 | 0 | 1 (8.3) |
| Metabolism and nutrition disorders | 3 (37.5) | 4 (50.0) | 3 (37.5) | 2 (33.3) | 3 (25.0) |
| Decreased appetite | 0 | 4 (50.0) | 2 (25.0) | 1 (16.7) | 0 |
| Hypoglycemia | 1 (12.5) | 0 | 1 (12.5) | 0 | 0 |
| Diabetic ketosis | 0 | 0 | 0 | 0 | 1 (8.3) |
| Cardiac disorders | 2 (25.0) | 1 (12.5) | 2 (25.0) | 1 (16.7) | 1 (8.3) |
| Atrioventricular block first degree | 0 | 0 | 2 (25.0) | 0 | 0 |
| Myocardial ischemia | 2 (25.0) | 0 | 0 | 0 | 0 |
| Ventricular extrasystoles | 0 | 1 (12.5) | 0 | 1 (16.7) | 0 |
| Arrhythmia | 1 (12.5) | 0 | 0 | 0 | 0 |
| Palpitations | 0 | 0 | 0 | 1 (16.7) | 1 (8.3) |
| Investigations | 0 | 1 (12.5) | 2 (25.0) | 0 | 3 (25.0) |
| Lipase increased | 0 | 0 | 2 (25.0) | 0 | 1 (8.3) |
| Hepatobiliary disorders | 1 (12.5) | 0 | 0 | 0 | 0 |
| Hepatic function abnormal | 1 (12.5) | 0 | 0 | 0 | 0 |

Data are presented as number of patients (%).
* By the Medical Dictionary for Regulatory Activities (version 24.0) system organ class and preferred term.

4.5 mg and 6.0 mg cohorts, respectively, compared with −1.98% (0.48) for dulaglutide and −0.87% (0.34) for placebo (Fig. 2a). Compared with placebo, a significant reduction in $HbA_{1c}$ levels was achieved at week 12 in patients receiving IBI362 in the 4.5 mg cohort (treatment difference −1.35% [90%CI − 2.30, −0.41], $P = 0.0209$) (Table S2). At week 12, 62.5% of patients treated with IBI362 achieved ≥1.5% reductions in $HbA_{1c}$ from baseline (33.3% with dulaglutide and 16.7% with placebo) (Fig. S3a), and nine patients treated with IBI362 achieved $HbA_{1c}$ level of 6.5% or lower (Fig. S3b). Of note, one super responder in patients receiving dulaglutide had $HbA_{1c}$ level dropped from 10.5% at baseline to 5.5% at week 12, while one non-responder in patients receiving IBI362 in the 6.0 mg cohort had $HbA_{1c}$ level increased from 7.6% at baseline to 8.4% at week 12 (Fig. S3b). Mean change from baseline to week 12 in FPG levels were −3.38 mmol/L (SE 0.49), −3.82 mmol/L (0.53) and −4.07 mmol/L (0.48) for patients receiving IBI362 in the 3.0 mg, 4.5 mg and 6.0 mg cohorts, respectively, compared with −3.85 mmol/L (0.55) for dulaglutide and −2.52 mmol/L (0.40) for placebo (Fig. 2b). Compared with placebo, a significant reduction in FPG levels was achieved at week 12 in patients receiving IBI362 in the 6.0 mg cohort (treatment difference −1.54 mmol/L [90%CI − 2.60, −0.49], $P = 0.0185$) (Table S2). At week 12, improvements on HOMA-β and HOMA-IR were observed in patients receiving IBI362 and dulaglutide (Fig. S4).

IBI362 treatment also reduced postprandial glucose levels (Fig. S5a). Mean percent change from baseline to week 12 in post-MTT glucose $AUC_{0-4\ h}$ were −37.0% (SE 5.2), −38.9% (6.2) and −43.9% (4.8) for patients receiving IBI362 in the 3.0 mg, 4.5 mg and 6.0 mg cohorts, respectively, compared with −41.7% (5.6) for

dulaglutide and −20.2% (4.1) for placebo (Fig. S5b). Compared with placebo, more reductions in mean SMBG levels were observed in patients receiving IBI362 and dulaglutide (Fig. S6). In addition, an increase in post-MTT insulin and C peptide excursion was observed at week 12 in patients receiving IBI362 and dulaglutide (Fig. S5, c, d). At week 12, post-MTT GLP-1 and OXM excursions were markedly suppressed in patients receiving IBI362, while reductions in fasting GLP-1 and OXM were evident in patients receiving IBI362 in the 4.5 mg and 6.0 mg cohorts (Fig. S5, e, f).

IBI362 treatment significantly reduced body weight. Mean percent change from baseline to week 12 in body weight were −0.9% (SE 1.3), −5.0% (1.3) and −5.4% (1.2) for patients receiving IBI362 in the 3.0 mg, 4.5 mg and 6.0 mg cohorts, respectively, compared with −0.9% (1.4) for dulaglutide and −1.1% (1.0) for placebo (Fig. 2c). Compared with placebo, significant reduction in body weight was achieved at week 12 in patients receiving IBI362 (treatment difference 4.5 mg: −4.0% [90%CI − 6.7, −1.2], $P = 0.0204$; 6.0 mg: −4.4% [−7.0, −1.7], $P = 0.0093$) (Table S2). Compared with dulaglutide, which provided glycaemic control with almost neutral effect on weight loss, IBI362 offered both clinically meaningful glycaemic control and weight loss benefits (Fig. 2d). Moreover, at week 12, more pronounced reductions in waist circumference, BMI, blood pressure, cholesterol and triglycerides levels were observed in patients receiving IBI362, compared with placebo (Fig. S7 and S8).

## Discussion

This 12-week randomised, placebo-controlled, active-referenced phase 1b study evaluated the safety, tolerability, pharmacokinetics and efficacy of IBI362, a balanced once-weekly GLP-1 and

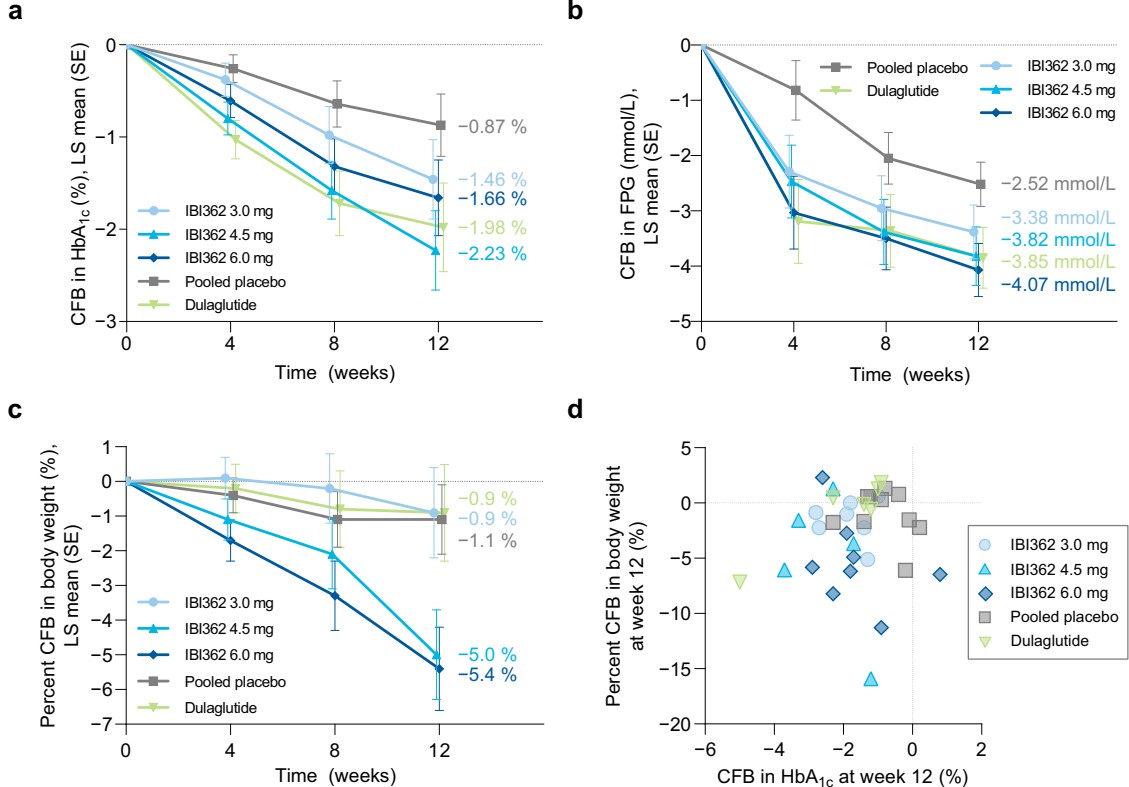

**Fig. 2 Change from baseline in HbA_{1c}, fasting plasma glucose and body weight over time and at week 12. a** CFB in HbA_{1c} levels over time. **b** CFB in FPG levels over time. **c** Percent CFB in body weight over time. **d** CFB in HbA_{1c} levels and percent CFB in body weight for each patient at week 12. Data in **a–c** are plotted as LS means ± SEM from an MMRM model, with LS means at week 12 shown alongside. IBI362 3.0 mg $n = 8$; IBI362 4.5 mg $n = 8$; IBI362 6.0 mg $n = 8$; Pooled placebo $n = 12$; Dulaglutide $n = 6$. CFB, change from baseline, FPG, fasting plasma glucose, HbA_{1c}, glycated haemoglobin A_{1c}, LS, least squares, MMRM, mixed-effect model for repeated measures, SE, standard error of the mean. Source data are provided as a Source Data file.

glucagon receptor dual agonist, in Chinese patients with T2D. With a favourable safety profile, IBI362 demonstrates both clinically meaningful glucose reduction and body weight loss effects, together with improvements in multiple metabolic parameters in patients with T2D.

The dose escalation regimens evaluated in this study were well tolerated. The overall safety profile was consistent with those of other GLP-1 receptor agonists and co-agonists[10,11,16]. Gastrointestinal symptoms of diarrhoea and nausea were most frequently reported in patients receiving IBI362 and dulaglutide while the incidence of vomiting was relatively low. Deceased appetite was mostly reported in patients receiving IBI362 in the 4.5 mg and 6.0 mg cohorts (Table 2). Given the small sample size and short duration of this study, the incidence and prevalence of gastrointestinal adverse events during long-term IBI362 treatment warrant further investigation.

GLP-1 receptor agonists were unequivocally associated with heart rate increase of varying magnitude, which, however, did not increase the risk of major adverse cardiac events in patients with T2D and high cardiovascular risk[17,18]. In light of the heart rate increase observed with direct infusion of glucagon in humans and the evidence that OXM increased intrinsic heart rate through the glucagon receptor in mice[19,20], more pronounced heart rate increase is expected with GLP-1 and glucagon receptor dual agonists. The heart rate increase observed in patients receiving IBI362 up to 10-15 beats per minute was numerically higher than other GLP-1 and glucagon receptor dual agonists[11,21]. However, the result was complicated by substantial variation potentially brought by small sample size and suboptimal measurement method (single measurement at each visit rather than ambulatory

blood pressure monitoring or 24-hour Holter monitoring used in other studies), both of which were limitation of this study. The effect of IBI362 on heart rate will be evaluated in larger population with long-term treatment in phase 2 studies.

While most patients treated with IBI362 may benefit from blood pressure reduction, blood pressure elevation accompanied by heart rate increase may reflect the increase in cardiac burden and be of safety concern. Two patients receiving IBI362 in the 6.0 mg cohort had potential concurrent increase in blood pressure and heart rate during the study and reported atrioventricular block first degree, albeit mild in severity, which necessitated careful monitoring of relevant safety issues in the future development of IBI362.

Several GLP-1 and glucagon dual agonists have shown either glycaemic control or weight loss benefit in various settings. Cotadutide, a once-daily GLP-1 and glucagon receptor dual agonist, achieved a 0.90–1.24% mean reduction (0.14% for placebo) in HbA_{1c} level in Japanese patients with T2D[22]. SAR425899, a once-weekly GLP-1 and glucagon receptor dual agonist, achieved placebo-adjusted mean HbA_{1c} level reductions up to 0.75% in patients with overweight or obesity and T2D[11]. JNJ-64565111, another once-weekly GLP-1 and glucagon receptor dual agonist showed encouraging weight loss but no improvement in glycaemic parameters in adults with T2D and obesity[23].

IBI362 has been designed to balance the activation of GLP-1 receptor and glucagon receptor in order to avoid the glucagon receptor-induced hyperglycaemia while maintaining the desired effects of HbA_{1c} reduction and body weight loss. Without a lead-in period, a relatively strong placebo effect was observed with

glycaemic efficacy outcomes in our study. Nevertheless, the effects of IBI362 on HbA_{1c}, FPG and postprandial glucose were generally comparable to those of dulaglutide. Notably, with a small sample size, the results of HbA_{1c} and FPG reductions were skewed by outliers. These factors, together with unbalanced baseline characteristics brought by in-cohort randomisation and short study duration, made the dose-response relationship less evident at this stage, which will be further investigated in phase 2 studies.

Another noteworthy finding of this study was the marked suppression of endogenous GLP-1 and OXM in patients treated with IBI362, in both fasting and postprandial settings. Suppression of postprandial GLP-1 secretion was also found in adults receiving cotadutide, a daily-dose GLP-1 and glucagon receptor dual agonist[10]. Furthermore, Roux-en-Y gastric bypass augments postprandial secretion of GLP-1 and OXM[24,25], and OXM may predict the effect of bariatric surgery on weight loss[26]. The suppression of endogenous GLP-1 and OXM in this study may reflect the effective agonism of the target receptors that underlies the observed efficacy on glycaemic control and weight loss. Further mechanistic investigation in broader ethnic groups with long-term treatment are needed to validate the effect of IBI362 on endogenous incretins and gastrointestinal hormones.

For GLP-1 and glucagon receptor dual agonists, the weight loss is expected from both the inhibition of increased satiety and reduced dietary intake induced by GLP-1 receptor agonism, and increase in energy expenditure by glucagon receptor stimulation[27,28]. IBI362 has demonstrated robust weight loss effect in Chinese adults with overweight or obesity[15]. With the same dose escalation regimens, the weight loss effect of IBI362 was further explored in Chinese patients with T2D, a population with significantly lower baseline body weight. In patients with comparable baseline body weight, daily-dose cotadutide achieved up to 3.3% reduction in body weight from baseline to day 50 in Japanese patients with T2D[22], and semaglutide 1.0 mg achieved body weight reduction of about 3.6% at week 12 in Chinese patients with T2D in SUSTAIN China study[29]. IBI362 achieved body weight reductions up to 5.4% from baseline to week 12 in patients with T2D (Fig. 2c). The weight loss, together with reductions in blood pressure, LDL cholesterol, triglycerides and transaminases levels, may confer comprehensive metabolic benefits to patients with T2D.

As a phase 1b study, the 12-week treatment period is relatively short for efficacy evaluation. Neither glycaemic parameters nor weight loss reached plateau at week 12. Furthermore, the small sample size and lack of lead-in period introduced variations and fluctuations to the results, precluding robust statistical analyses and conclusion. A phase 2 study had been launched to evaluate the efficacy and safety of IBI362 in Chinese patients with T2D in larger, multicentre, long-term settings.

In summary, this study demonstrated that IBI362 was well tolerated with an overall favourable safety profile in Chinese patients with T2D. IBI362 showed clinically meaningful efficacy on secondary outcomes of glycaemic control and the pre-specified exploratory outcome of weight loss. Taken together, our data indicate that IBI362 holds potential to treat T2D and other metabolic disorders.

## Methods

**Study design.** This randomised, placebo-controlled, dose-escalation, multiple-ascending-dose study, with dulaglutide as active reference, was designed to explore the optimal dosing regimens and assess the safety, tolerability, pharmacokinetics and efficacy of IBI362 in Chinese patients with T2D. Patients were enroled from nine study centres in China. The study was done in accordance with local laws, the International Conference on Harmonization Good Clinical Practice guidelines, and the ethical principles outlined in the Declaration of Helsinki. The clinical study protocol, the protocol amendment and informed consent forms were approved by the following ethics committees: the ethics committee of Henan University of

Science and Technology First Affiliated Hospital (Jinghua Division); the ethics committee of Henan University of Science and Technology First Affiliated Hospital (Kaiyuan Division); the ethics committee of Jinan Central Hospital; the ethics committee of Pingxiang People's Hospital; the ethics committee of China-Japan Friendship Hospital; the ethics committee of Bengbu Medical College First Affiliated Hospital; the ethics committee of Guizhou Medical University Affiliated Hospital; the ethics committee of Shanxi Medical University First Affiliated Hospital; the ethics committee of Tonghua Central Hospital. This study is registered with ClinicalTrials.gov, number NCT04466904.

**Patients.** Patients (aged 18–75) who had been diagnosed with T2D for at least six months that was inadequately controlled with diet and exercise alone or with stable metformin therapy (glycated haemoglobin A_{1c} [HbA_{1c}] 7.5–11.0%, both inclusive) and a body-mass index (BMI) of 20–35 kg/m$^2$ (both inclusive) were eligible for this study. All patients provided written informed consent before study entry.

**Randomisation and masking.** An interactive web-response system-generated identification numbers were used to randomly assign eligible patients 8:4:2 to receive IBI362, placebo or open-label dulaglutide in each cohort. The randomisation schedule was generated by an in-house statistician who was not involved in the clinical operation of the study. IBI362 and placebo were identically labelled and indistinguishable in appearance. As such, the patients, investigators, study site personnel involved in treating and assessing patients and sponsor personnel in each cohort were masked to IBI362 and placebo allocation.

**Procedures.** The study included a 3-week screening period, a 12-week treatment period and an 8-week safety follow-up period. During the 12-week treatment period, IBI362 or placebo was subcutaneously administered once weekly with one of the three dose-escalation schedules: 1.0–2.0–3.0 mg for the 3.0 mg cohort, 1.5–3.0–4.5 mg for the 4.5 mg cohort and 2.0–4.0–6.0 mg for the 6.0 mg cohort (each dose level administered for 4 weeks). The 6.0 mg cohort started dosing after tolerability evaluation of 1.5 mg dosing in the 4.5 mg cohort. The internal unblinded safety monitoring group was responsible for the safety monitoring. Dulaglutide was subcutaneously administered once weekly at a fixed dose of 1.5 mg. Patients who were on metformin prior to the enrolment of this trial were required to continue their previous dose of metformin throughout the trial. Investigators and authorized study personnel in each centre recorded the study data into the electronic case report forms (eCRFs), which were reviewed and validated by clinical research associates. The data in the eCRFs were submitted to the electronic data capture database.

HbA_{1c} (2200101, Bio-Rad, USA), blood glucose (05168791190, Roche Diagnostics, Germany), insulin (12017547122, Roche Diagnostics, Germany), C peptide (03184897190, Roche Diagnostics, Germany), total GLP-1 (GLP-1 [7–36] amide and its metabolite, GLP-1 [9–36] amide, 10-1278-01, Mercodia AB, Sweden) and OXM (AL-139, AnshLabs, USA) tests were performed according to the manufactures' instruction manuals in a central laboratory (KingMed Diagnostics, Guangzhou, China). HbA_{1c} was measured using a Bio-rad D-100 Haemoglobin Testing System (Bio-rad). Blood glucose was measured using a Roche Cobas c702 analyser (Roche Diagnostics). Insulin and C peptide were measured using a Roche Cobas c602 analyser (Roche Diagnostics). GLP-1 and OXM were measured using a Tecan Infinite M Plex microplate reader (TECAN).

**Outcomes.** Primary outcomes of the study were the safety and tolerability of IBI362. Safety assessment included adverse events, physical examinations, laboratory tests (including lipase, amylase and calcitonin level measures), vital signs and 12-lead electrocardiogram. Adverse events were categorized according to the Medical Dictionary for Regulatory Activities system organ classes (SOCs) and preferred terms. Treatment-emergent adverse events (TEAEs) were monitored from the time of informed consent to the safety follow-up period. The severity of TEAEs (mild, moderate or severe) and the association between an event and the study drug were assessed by investigators based on pre-specified criteria.

Secondary outcomes included pharmacokinetics, assessed by maximum observed plasma concentration (C_{max}), time at which C_{max} was observed (T_{max}), terminal elimination half-life (t_{½}), and area under the curve from time zero to 168 h after the first dose (AUC_{0–168 h}) and immunogenicity, assessed by titres of anti-drug antibodies (ADAs) and neutralizing antibodies. Standard non-compartmental pharmacokinetics methods were used to analyze IBI362 plasma concentration data using Pkanalix 2020R1 (Lixoft, Antony, France). Secondary efficacy outcomes included change from baseline to week 12 in HbA_{1c}, fasting plasma glucose (FPG), fasting insulin levels, change from baseline to week 12 in post-mixed-meal tolerance test (post-MTT) glucose, insulin, C peptide, GLP-1 and OXM levels, as well as seven-point self-measured blood glucose (SMBG) profile.

Exploratory outcomes included change from baseline in body weight, waist circumference, body-mass index (BMI) and change in homeostatic model assessment of β-cell function (HOMA-β) and insulin resistance (HOMA-IR).

**Statistical analysis.** The sample size was determined based on sample sizes commonly used in previous early phase studies of medications in the same drug class[10,11]. Eight patients received IBI362 in each cohort to provide preliminary

safety, tolerability and pharmacokinetics data of IBI362, while a total of six patients received open-label dulaglutide to provide a reference for safety and efficacy. Patients receiving placebo ($n = 12$) and dulaglutide ($n = 6$) were each separately pooled across the three cohorts for analysis.

Safety analyses were done in the safety population (defined as all patients who received at least one dose of the study treatment). Pharmacokinetic analyses were done in the pharmacokinetic population (defined as all patients who received at least one dose of IBI362 and had at least one valid plasma concentration assessment). Efficacy analyses were done in the efficacy population (defined as all patients who received at least one dose of the study treatment and had baseline and at least one post-baseline $HbA_{1c}$ assessment). Immunogenicity analyses were done in the immunogenicity population (defined as all patients who received at least one dose of the study treatment and had at least one post-baseline ADA result).

The analyses for change from baseline in $HbA_{1c}$, FPG, post-MTT glucose $AUC_{0-4\,h}$ and body weight were performed using the mixed-effect model for repeated measures (MMRM) with least-squares means and standard error provided. The MMRM included the corresponding baseline value, treatment, visit and treatment-by-visit as fixed effects and patient as a random effect. Mean change or mean percent change from baseline were analysed using a restricted maximum likelihood (REML)-based repeated measures approach in combination with the Newton Raphson Algorithm. The fix effect of visit was fitted as a repeated effect and an unstructured covariance matrix was used. In case of non-converge, matrix like Variance Components and Compound Symmetry will be tested in a subsequent order until model-convergence is achieved. The Kenward-Roger degree of freedom was used in estimating fixed effects. No missing data imputation was conducted. For $HbA_{1c}$, FPG and body weight, point estimate of treatment difference versus placebo and 90% CI were provided for each IBI362 dose group and the dulaglutide group, with nominal p values.

Summaries for other efficacy and safety variables included descriptive statistics for continuous measures (mean and standard deviation or standard error) and for categorical measures (sample size, frequency and percentages).

All statistical analyses were done using SAS version 9.4.

**Reporting summary**. Further information on research design is available in the Nature Research Reporting Summary linked to this article.

## Data availability
Individual de-identified patient data underlying the results reported in this article will be made available to investigators whose proposed use of the data has been approved by the corresponding authors (W.Y. or L.Q.), beginning 9 months and ending 36 months following article publication. To gain access, data requestors must enter into a data access agreement with Innovent Biologics. The clinical study protocol and statistical analysis plan are provided as Supplementary Note 1 and Supplementary Note 2, respectively. Source data are provided with this paper.

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

## Acknowledgements
This study was sponsored by Innovent Biologics, Inc. The sponsor was involved in the study design, data collection, data review, data analysis, and drafting of the manuscript. We thank all the patients, investigators and study site staff who were involved in the conduct of this trial.

## Author contributions
W.Y., L.Q., M.L. and T.Y. designed the study. W.Y., H.J., S.P., Y.Z. and T.Y. did the trial and collected the data. L.L., L.F. and B.S. analysed the data. L.Q., M.L., H.D., T.Y., L.L., L.F., B.S., H.H.-Z. and Q.M. interpreted the data. H.H.-Z. and Q.M. wrote the manuscript. All authors had full access to all the data in the study and had critically reviewed the manuscript and approved the final manuscript. All authors vouch for data accuracy and fidelity to the protocol.

## Competing interests
W.Y., H.J., S.P. and Y.Z. received research funding from Innovent Biologics, Inc., during the conduct of the study. L.Q., M.L., H.D., T.Y., L.L., L.F., B.S., H.H.-Z. and Q.M. were employees of Innovent Biologics, Inc.
