## [Peer Review File · Nature Communications]

Title: A phase 1b randomised controlled trial of IBI362 (LY3305677) in Chinese patients with type 2 diabetesREVIEWER COMMENTS

Reviewer #1 (Remarks to the Author):

Summary

IBI362 (LY3305677) is a novel weekly-dose glucagon-like peptide-1 and glucagon receptor dual agonist being developed for the treatment of obesity and type 2 diabetes. It is a long-acting synthetic peptide analogue of mammalian oxyntomodulin with a fatty-acyl moiety to extend the half-life.

This manuscript described a 12-week randomized, placebo-controlled, multiple ascending dose phase 1b study (NCT04466904) that evaluated the safety, tolerability, pharmacokinetics (PK) and efficacy of IBI362 in Chinese participants with type 2 diabetes (T2D). Forty-three participants with T2D, suboptimally controlled on diet alone or with stable metformin therapy, were randomized in 8:4:2 ratio into 3 cohorts to receive multiple ascending doses of IBI362 up-titrated to 3 mg, 4.5 mg or 6 mg, placebo or open-label weekly dulaglutide 1.5 mg, for 12 weeks, with an 8-week safety follow-up period. Primary endpoints were safety and tolerability of IBI362. Secondary endpoints included changes from baseline to week 12 in glycated hemoglobin (HbA1c), fasting plasma glucose (FPG), and post-mixed meal tolerance test (MTT) glucose levels, body weights, PK and immunogenicity of IBI362. Three participants in the IBI362 active treatment arm were terminated in the trial: one due to adverse event, one due to COVID-19 and a third for other unspecified reason. Two serious adverse events, and both were cardiac-related, were reported in the IBI362 3.0 mg cohort, with one deemed unrelated to drug by the study investigators. All treatment-emergent adverse events (TEAEs) were mild to moderate in severity, and were mostly gastrointestinal disorders and decreased appetite. Dose-dependent reductions were observed in HbA1c, FPG and post-MTT glucose levels, along with body weight loss in the three cohorts, when compared with placebo or the positive control dulaglutide. The authors concluded that IBI362 demonstrated a favourable safety and tolerability profile, and clinically meaningful reductions in HbA1c, glucose levels and body weight in Chinese participants with T2D and suboptimal control by diet alone or with metformin.

Critique

This multicentre randomized control trial, which included a 3-week screening period, a 12-week treatment period and an 8-week safety follow-up period, was conducted in nine study centres in China. The sample size in this trial was small and empirically determined to provide adequate safety, tolerability, and PK data. The inclusion and exclusion criteria, primary and secondary endpoints were clearly stated in the study protocol. Of the 42 participants randomized, 38 completed the study and 37 completed treatment. The data collected were valid and appropriate statistical analyses were performed.

Safety and tolerability data

With respect to the primary endpoints of safety and tolerability, most of the TEAEs were gastrointestinal (GI) disorders that were mild to moderate in severity, and appeared to increase with the two higher

doses of IBI362 (4.5 and 6 mg). It would be helpful to provide the time course of the GI adverse events to ascertain whether they occurred during the up-titration phase and whether the symptoms persisted during the 12 weeks. Only one participant in the IBI362 4.5 mg cohort withdrew from the study due to decreased appetite.

The cardiovascular safety profile of IBI362 reported in this manuscript raised some safety concerns and will be addressed in greater detail herein.

Cardiovascular disorders were reported in 5 participants receiving IBI362 compared with one in each of the placebo and dulaglutide treatment arms. There were 2 serious adverse events (SAEs), both were myocardial ischemia reported in two participants in the IBI362 3 mg cohort, with one also suffered from cardiac arrhythmia with premature ventricular contractions. Both SAEs were judged by the study investigators as unrelated to IBI362 treatment.

GLP-1 receptor agonists are known to increase heart rate by several beats per minute (BPM), presumably due to activation of GLP-1 receptors in the sinoatrial node of the heart. However, several cardiovascular outcome studies on long-acting GLP-1 agonists (LEADER trial with liraglutide, SUSTAIN trial with semaglutide and REWIND trial with dulaglutide) in people with T2D and established cardiovascular disease were associated with reduction in major adverse cardiovascular events, such as nonfatal myocardial infarction, stroke or cardiovascular deaths, rather than harm. The CV benefits of GLP-1 receptor agonists could at least partially attributable to their blood pressure lowering and anti-atherosclerotic properties despite a small increase in heart rate. While a more pronounced increase in heart rate is anticipated in dual GLP-1 and glucagon receptor agonists based on published human and rodent data, a persistent heart rate increase of up to 15 BPM from baseline in participants in the IBI362 6 mg cohort (Fig. S1) is of safety concern. In contrast, the mean increase in heart rate, systolic and diastolic BP values reported with a phase 2a study with another dual GLP-1/glucagon receptor agonist cotatdutide were +7.1 BPM, -2.8 and +1.16 mm Hg, respectively, after 49 days of drug exposure (Parker VER et al. JCEM 2021;105(3):803-20).

An increase in heart rate may potentially increase systolic and diastolic blood pressure (BP) as well as cardiovascular disease risk in people with T2D. While the authors reported reduction in mean blood pressure with IBI362 treatment, there were 5 outliers (Fig. S5C) with vastly increased systolic (one greater than 30 mm Hg) and diastolic BP (one with greater than 20 mm Hg) values in the IBI362 6 mg cohort. It would be helpful to provide the absolute BP values of these participants. This also raised the question of relationship to the 2 CV events reported in Table 2. Were these the same or different participants who were found to have first degree AV block?

Ambulatory BP and heart rate measurements, which would provide assurance on greater accuracy, were not reported in this study.

Efficacy data

IBI362 treatment reduced HbA1c and FPG values in a dose-dependent manner, similar to those in the

dulaglutide treatment arm. Fig. 2C showed much greater body weight reductions with the 4.5 and 6 mg cohorts (-5% and -5.4%) compared to dulaglutide (-0.9%) or placebo (-1.1%). IBI362 treatment also showed dose-dependent reductions in waist circumference and BMI values (Fig. S5A and S5B).

There was a non-responder in the IBI362 6 mg cohort, with a reported increase in HbA1c from 7.6% at baseline to 8.4% at week 12 despite a 6.5% body weight reduction (Figs. 2D and S3B).

With respect to secondary efficacy endpoints, IBI362 treatment was associated with reduction in post-prandial glucose levels, increased post-MTT insulin excursion, C-peptide levels, and oral glucose insulin sensitivity, similar to those with dulaglutide treatment. More detailed assessment on glucose excursions using seven-point glucose measurements or continuous glucose monitoring would be helpful to provide further insights into the glycemic effects of IBI362 dual agonist. Nonetheless, the available data allay the concern of glucagon's stimulatory effect on hyperglycemia. Unfortunately there were no data on HOMA-beta or HOMA-IR to better assess changes in insulin resistance and insulin sensitivity.

IBI362 treatment was associated with a modest reduction in total cholesterol, LDL-cholesterol and triglyceride levels similar to dulaglutide treatment (Fig. S6). ALT and AST liver enzymes generally decreased slightly from baseline to week 12 with IBI362 treatment, with one participant in the 3 mg cohort showing a transient increase in ALT level that peaked at week 4.

An unexpected finding was the significant suppression rather than augmentation of endogenous GLP-1 and oxyntomodulin levels in fasting and post-prandial states with all 3 doses of IBI362 treatment (Fig S4F and S4G). The authors posited "a feedback inhibition mechanism of gastrointestinal hormone pathways". This observation clearly requires further investigation and confirmation in a larger and different patient population other than Chinese.

The rationale for pursuing the development of a dual GLP-1 and glucagon receptor co-agonist is to exploit the stimulatory effect of glucagon on energy expenditure, and which might lead to greater weight loss than what can be achieved through central appetite suppression with a GLP-1 receptor agonist alone. Glucagon infusion in humans and rodents increased oxygen consumption, which is presumably driven by its thermogenic effects on the liver and adipose tissue. This study did not investigate the effect of IBI362 treatment on energy expenditure using doubly labelled water, respirometry, oxygen consumption, or even with activity monitors, which represents a missed opportunity. Glucagon may exert catabolic effect on amino acid and protein metabolism, and could potentially have undesirable effect on muscle and lean body mass. Hence body composition measurements with dual GLP-1 and glucagon receptor co-agonist are useful tools to assess lean muscle and body fat mass.

Significance

Overall this small, short-term phase 1b study with IBI362, a novel weekly GLP-1 and glucagon receptor dual agonist, demonstrated an acceptable safety and efficacy profile. IBI362 once weekly treatment also led to fairly robust improvements in glycemic control and body weight loss and these effects appear to

greater than those reported in a 54-week phase 2b randomized trial on cotadutide, a daily GLP-1 and glucagon receptor dual agonist under development (Nahra R et al. Diabetes Care 2021;44:1433-42). The more pronounced increase in heart rate expected in a dual agonist, which has also been reported by other dual GLP-1 and glucagon receptor agonists under development, remains a safety concern. The magnitude of glucose-lowering and weight loss benefits of IBI362 treatment reported in this manuscript is similar, and perhaps marginally better than those of such long-acting GLP-1 receptor agonists as liraglutide and semaglutide, both of which are currently approved for diabetes and obesity treatment. Current data warrant further evaluation of this novel dual agonist in a phase 2 trial, which will provide more clarity on its potential development as a treatment option for obesity and T2D.

Reviewer #2 (Remarks to the Author):

Summary: This paper describes a randomized, placebo-controlled, multiple ascending dose phase 1b study to evaluate the safety, tolerability, pharmacokinetics, and efficacy of a novel weekly-dose GLP-1 and glucagon receptor dual agonist (IBI362) for the treatment of type 2 diabetes and obesity. This study compares the performance of three different ascending dose regimens for IBI362 to placebo and an active reference treatment (dulaglutide). The study concluded that the safety profile for all three dose regimens of IBI362 to be acceptable, that the efficacy of IBI362 for glycemic control was comparable to the active reference dulaglutide, and that reductions in body weight were greater for IBI362 compared to the active reference dulaglutide. Overall, this paper provides a meaningful contribution to the field regarding treatments for diabetes and obesity. However, there are a few minor issues that should be clarified prior to acceptance for publication:

1. The inclusion criteria for this study restrict the sample to patients whose diabetes was inadequately controlled with diet and exercise alone or with stable metformin therapy. For those patients who were using metformin therapy prior to enrollment in the study, was metformin therapy stopped prior to enrollment in the study, or were the study treatments added to the existing metformin therapy? If metformin therapy was stopped prior to enrollment in the study, would the 3-week screening period be considered a sufficient “wash-out period” to ensure that the effects of the metformin therapy would not influence the results?
2. In the “Randomization and masking” section, it is stated that the active reference dulaglutide group was open-label (line 115). However, it also states that “the study drugs and placebo were identically labeled and indistinguishable in appearance” (lines 117 – 118). These two statements seem to be contradictory. Please clarify whether it was known who was in the dulaglutide group, or if this group was masked in a similar manner as the IBI362 and placebo groups.
3. On lines 163 – 164 in the “Statistical analysis” section, it states that “the sample size in this study was empirically determined to provide adequate safety, tolerability and pharmacokinetics data.” It is unclear

what this statement means. Please clarify how the sample size was determined for this study.

4. On line 165 in the “Statistical analysis” section, it states that the placebo and dulaglutide groups were pooled for analysis. However, this statement seems inconsistent with what is presented in the results; all tables and figures appear to present results separately for the placebo and dulaglutide groups. Please clarify whether these groups were pooled for analyses, and if so then for which analyses.

5. On line 201 in the Results section, it is stated that of the 42 patients in the safety and efficacy population, 37 patients completed the treatment. For patients who discontinued the study treatment prior to completing or discontinuing the study, were outcome measures collected for these patients following treatment discontinuation, and if so were data following treatment discontinuation included in statistical analyses (i.e., for intention-to-treat analyses)?

6. For the tables that present treatment emergent adverse events (Table 2 and Supplementary Table 2), do the numbers in the table correspond to the number of events, or the number of patients who experienced at least one event?

7. For Supplementary Table 1: Why is the terminal elimination half-life outcome ($t_{1/2}$) not included in this table?

Reviewer #3 (Remarks to the Author):

Jiang report on a phase 1 b clinical trial with a novel dual GLP-1/glucagon co-agonist in Chinese patients with type 2 diabetes. This compound (IBI362/LY3305677) is used with once-weekly injections. Compounds addressing several pancreatic or intestinal hormone receptors are an important area of research to improve glycaemic control and body weight in subjects with type 2 diabetes and obesity. The present study reports substantial improvements in glycated haemoglobin and body weight, indicating some therapeutic potential for the compound studied. However, results with placebo treatment indicate significant changes vs. baseline. This complicates the interpretation. The overall number of patients studied was low (6-8 per group with active drugs). This explains some of the problems mentioned in detail below.

Major points:

(1) The main outcomes should be reported as placebo-corrected effects. If the statistical analysis plan requires to report the effects as they occurred, the placebo-corrected values should be added as a meaningful post-hoc analysis. The fact that there were no clear dose-response relationships should be explained.

(2) The rationale behind stimulating glucagon as well as GLP-1 receptors is additional weight loss (increases in energy expenditure due to glucagon agonism). It is unclear why weight reduction was treated as exploratory endpoint.

(3) Line 165: The statement that “patients receiving placebo or dulaglutide were pooled for analysis” is unclear and may be misleading. Placebo and dulaglutide should not be lead to similar effects.

(4) The ECG abnormalities did newly occur during treatment with IBI362/LY3305677?

(5) The use of modelling to estimate insulin sensitivity probably is misleading when using an agent that likely decelerates gastric emptying: This should result in lower glucose increments with lower insulin response, a combination which will be interpreted as an increased insulin sensitivity. See also discussion (lines 339-341).

Minor points:

(a) Line 40 ff. and elsewhere: Since nausea and vomiting are most likely caused by a direct interaction of the agent with the brain stem and does not involve functional changes in the GI tract, the term “gastrointestinal disorders” should be replaced by “gastrointestinal symptoms (nausea, vomiting and diarrhoea)”. The abstract should present numbers regarding weight loss (line 46). The low frequency of vomiting is remarkable.

(b) Line 68: typo: Addictive should be additive

(c) The introduction mentions a previous multiple ascending dose study with this agent in Chinese adults. The rationale for performing a second study should be mentioned along these lines. The identification of optimal dosing regimens is usually the main objective in phase 2.

(d) What is the purpose of measuring amidated GLP-1 (line 137)? This can still be active (7-36 amide) or inactive (9-36 amide), so why not measure total or intact GLP-1?

(e) Why did dulaglutide not change weight?

(f) The detailed report of results (lines 331-338) should be moved to the results section

Michael A. Nauck

REVIEWER COMMENTS

Reviewer #1 (Remarks to the Author):

Summary

IBI362 (LY3305677) is a novel weekly-dose glucagon-like peptide-1 and glucagon receptor dual agonist being developed for the treatment of obesity and type 2 diabetes. It is a long-acting synthetic peptide analogue of mammalian oxyntomodulin with a fatty-acyl moiety to extend the half-life.

This manuscript described a 12-week randomized, placebo-controlled, multiple ascending dose phase 1b study (NCT04466904) that evaluated the safety, tolerability, pharmacokinetics (PK) and efficacy of IBI362 in Chinese participants with type 2 diabetes (T2D). Forty-three participants with T2D, suboptimally controlled on diet alone or with stable metformin therapy, were randomized in 8:4:2 ratio into 3 cohorts to receive multiple ascending doses of IBI362 up-titrated to 3 mg, 4.5 mg or 6 mg, placebo or open-label weekly dulaglutide 1.5 mg, for 12 weeks, with an 8-week safety follow-up period. Primary endpoints were safety and tolerability of IBI362.

Secondary endpoints included changes from baseline to week 12 in glycated hemoglobin (HbA1c), fasting plasma glucose (FPG), and post-mixed meal tolerance test (MTT) glucose levels, body weights, PK and immunogenicity of IBI362. Three participants in the IBI362 active treatment arm were terminated in the trial: one due to adverse event, one due to COVID-19 and a third for other unspecified reason. Two serious adverse events, and both were cardiac-related, were reported in the IBI362 3.0 mg cohort, with one deemed unrelated to drug by the study investigators. All treatment-emergent adverse events (TEAEs) were mild to moderate in severity, and were mostly gastrointestinal disorders and decreased appetite. Dose-dependent reductions were observed in HbA1c, FPG and post-MTT glucose levels, along with body weight loss in the three cohorts, when compared with placebo or the positive control dulaglutide. The authors concluded that IBI362 demonstrated a favourable safety and tolerability profile, and clinically meaningful reductions in HbA1c, glucose levels and body weight in Chinese participants with T2D and suboptimal control by diet alone or with metformin.

Critique

This multicentre randomized control trial, which included a 3-week screening period, a 12-week treatment period and an 8-week safety follow-up period, was conducted in nine study centres in China. The sample size in this trial was small and empirically determined to provide adequate safety, tolerability, and PK data. The inclusion and exclusion criteria, primary and secondary endpoints were clearly stated in the study protocol. Of the 42 participants randomized, 38 completed the study and 37 completed treatment. The data collected were valid and appropriate statistical analyses were performed.

Safety and tolerability data

With respect to the primary endpoints of safety and tolerability, most of the TEAEs were gastrointestinal (GI) disorders that were mild to moderate in severity, and appeared to increase with the two higher doses of IBI362 (4.5 and 6 mg). It would be helpful to provide the time course of the GI adverse events to ascertain whether they occurred during the up-titration phase and whether the symptoms persisted during the 12 weeks. Only one participant in the IBI362 4.5 mg cohort withdrew from the study due to decreased appetite.

Response: We thank the reviewer for the comments. As with all the GLP-1 receptor agonists and dual agonists, gastrointestinal adverse events were observed in this study, mostly mild in severity. We re-analysed the by-week combined incidence of diarrhoea, nausea and vomiting during the treatment. These events occurred more frequently in the IBI362 4.5 mg and 6.0 mg groups. In the IBI362 6.0 mg group, slightly more patients reported gastrointestinal adverse events in the up-titration phases of 4.0 mg and 6.0 mg (Fig. R1). However, due to small sample size and relatively low incidence of these events, no conclusions could be made at this stage.

Fig. R1: By-week combined incidence of diarrhoea, nausea and vomiting

Gastrointestinal adverse events (diarrhoea, nausea and vomiting) were mostly transient. The median time to resolution of gastrointestinal adverse events were 1 day (interquartile range 1-3) for IBI362 3.0 mg group (four events), 3.5 days (1-5) for IBI362 4.5 mg group (10 events), 4 days (2-6) for IBI362 6.0 mg group (11 events), 3 days (interquartile range 2-4) for dulaglutide group (13 events). We added a description in the Results (Lines 235-237).

Gastrointestinal adverse events were well tolerated among patients receiving IBI362. No dose adjustment was made during the study. One patient in the IBI362 4.5 mg group experienced mild decreased appetite two days after the first dose and voluntarily quit the study after receiving two doses of IBI362 1.5 mg.

The cardiovascular safety profile of IBI362 reported in this manuscript raised some safety concerns and will be addressed in greater detail herein.

Cardiovascular disorders were reported in 5 participants receiving IBI362 compared with one in each of the placebo and dulaglutide treatment arms. There were 2 serious adverse events (SAEs), both were myocardial ischemia reported in two participants in the IBI362 3 mg cohort, with one also suffered from cardiac arrhythmia with premature ventricular contractions. Both SAEs were judged by the study investigators as unrelated to IBI362 treatment.

GLP-1 receptor agonists are known to increase heart rate by several beats per minute (BPM), presumably due to activation of GLP-1 receptors in the sinoatrial node of the heart. However, several cardiovascular outcome studies on long-acting GLP-1 agonists (LEADER trial with liraglutide, SUSTAIN trial with semaglutide and REWIND trial with dulaglutide) in people with T2D and established cardiovascular disease were associated with reduction in major adverse cardiovascular events, such as nonfatal myocardial infarction, stroke or cardiovascular deaths, rather than harm. The CV benefits of GLP-1 receptor agonists could at least partially attributable to their blood pressure lowering and anti-atherosclerotic properties despite a small increase in heart rate. While a more pronounced increase in heart rate is anticipated in dual GLP-1 and glucagon receptor agonists based on published human and rodent data, a persistent heart rate increase of up to 15 BPM from baseline in participants in the IBI362 6 mg cohort (Fig. S1) is of safety concern. In contrast, the mean increase in heart rate, systolic and diastolic BP values reported with a phase 2a study with another dual GLP-1/glucagon receptor agonist cotatdutide were +7.1 BPM, -2.8 and +1.16 mm Hg, respectively, after 49 days of drug exposure (Parker VER et al. JCEM 2021;105(3):803-20).

An increase in heart rate may potentially increase systolic and diastolic blood pressure (BP) as well as cardiovascular disease risk in people with T2D. While the authors reported reduction in mean blood pressure with IBI362 treatment, there were 5 outliers (Fig. S5C) with vastly increased systolic (one greater than 30 mm Hg) and diastolic BP (one with greater than 20 mm Hg) values in the IBI362 6 mg cohort. It would be helpful to provide the absolute BP values of these participants. This also raised the question of relationship to the 2 CV events reported in Table 2. Were these the same or different participants who were found to have first degree AV block?

***Response:** We thank the reviewer for the comments regarding the cardiac disorders, heart rate increase and blood pressure changes.*

Both serious adverse events of myocardial ischemia were reported after the last dose, during the safety follow-up period. One patient in the IBI362 3.0 mg group experienced asymptomatic premature ventricular contractions at week 8, revealed by electrocardiogram. The event persisted through week 20, when the electrocardiogram returned to normal. The event was recorded by the investigator as cardiac arrhythmia,

moderate in severity. The patient was diagnosed with ischemic heart disease, leading to hospitalization at week 15 (3 weeks after the last dose). The event was recoded as myocardial ischemia, unrelated to the study drug as judged by the investigator.

Another patient in the IBI362 3.0 mg group reported abnormal Q waves in the electrocardiogram, accompanied by chest tightness and shortness of breath. The patient was diagnosed with coronary heart disease, and reported serious adverse event of myocardial ischemia, unrelated to the study drug as judged by the investigator.

Of note, both adverse events were reported in the safety follow-up period and in the IBI362 3.0 mg group. While the causal relationship could not be concluded in this study with limited sample size and short study duration, the cardiac events should be carefully monitored and managed in the later stage development of IBI362.

*In terms of heart rate, we agree with the reviewer that persistent heart rate increase of 10-15 bpm is alarming and of safety concern. The magnitude of heart rate increase was similar to those observed in the multiple ascending dose phase 1b study of IBI362 in Chinese participants with overweight or obesity, which adopted the same dose regimens of IBI362¹. We also noted from publications of several GLP-1 and glucagon receptor dual agonists that typically 6-10 bpm increase in heart/pulse rate was observed^{2, 3, 4, 5, 6}, although prominent heart rate increases were observed in some early-phase studies^{5, 7}. In these studies, pulse rate or heart rate were mostly measured by using ambulatory blood pressure monitoring (ABPM) or 24-hour Holter monitoring, both of which, as the reviewer mentioned, added significant accuracy to the results. While it seems indisputable that glucagon receptor dual agonism would lead to further increase in heart rate, compared with GLP-1 receptor agonists, the increase in heart rate observed in our study would be appropriately interpreted as early-phase signals complicated by small sample size and suboptimal measurement method (single measurement by electrocardiogram). The variations in heart rate changes over time among patients were substantial (**Fig. R2**). Moreover, in several patients (in IBI362 as well as in dulaglutide and placebo groups), prominent heart rate increase was observed one week after the first dose and persisted throughout the treatment period, which may be attributed to inaccurate measurement of heart rate at baseline (before the first dose) (**Fig. R2**). The effect of IBI362 on heart rate will be*

evaluated in larger population with long-term treatment, in two phase 2 studies. We revised the discussion of heart rate increase in the Discussion (Lines 343-350).

Fig. R2: Heart rate change over time for individual patient
CFB = change from baseline.

Systolic blood pressure tended to decrease over time in patients receiving IBI362, although variation was evident (Fig. R3). At week 12, most patients treated with IBI362 had reductions in systolic blood pressure (Fig. R4). Among patients with blood pressure elevation at week 12, two in the IBI362 6.0 mg group reported adverse events of first degree atrioventricular block during the study, with all events mild in severity (Fig. R4, patient 1 and 2).

Fig. R3: Blood pressure changes over time
 CFB = change from baseline; SD = standard deviation.

Fig. R4: Changes from baseline to week 12 in heart rate and blood pressure for each patient.
 CFB = change from baseline.

We took a detailed examination of the heart rate and blood pressure changes over time for these two patients. Patient 1 had decrease in heart rate for most time during the treatment. The patient reported a transient episode of the event shortly after the first dose. The second episode started at week 4 when heart rate and blood pressure

were both on the rise, but not exceeding baseline values. The third episode reported by patient 1 started at week 8 when heart rate was decreasing despite a slight increase in the blood pressure (Fig. R5).

Fig. R5: Change in heart rate and blood pressure over time for patients reporting first degree atrioventricular block (shaded areas indicate the duration of adverse event of first degree atrioventricular block)

DBP = diastolic blood pressure; HR = heart rate; SBP = systolic blood pressure.

Patient 2 had heart rate increased during the treatment, peaking at week 5. The patient reported the first episode at week 6, when blood pressure was relatively stable despite an increase in heart rate from baseline by 10 bpm. The second episode of this patient started at week 8, when blood pressure was on the rise, accompanied by an elevated but decreasing heart rate (Fig. R5).

While most patients treated with IBI362 may benefit from blood pressure reduction, we agreed with the reviewer that blood pressure elevation accompanied by heart rate increase may reflect the increase in cardiac burden and is of safety concern, which should be monitored with caution in future development of IBI362. We added a discussion in Lines 351-357.

Ambulatory BP and heart rate measurements, which would provide assurance on greater accuracy, were not reported in this study.

Response: As discussed above, the lack of use of ambulatory blood pressure monitoring in this study added complication in the interpretation of the results observed. The safety signals observed in this study will be evaluated in phase 2 and 3 studies with more accurate measurement of relevant indicators.

Efficacy data

IBI362 treatment reduced HbA1c and FPG values in a dose-dependent manner,

similar to those in the dulaglutide treatment arm. Fig. 2C showed much greater body weight reductions with the 4.5 and 6 mg cohorts (-5% and -5.4%) compared to dulaglutide (-0.9%) or placebo (-1.1%). IBI362 treatment also showed dose-dependent reductions in waist circumference and BMI values (Fig. S5A and S5B).

There was a non-responder in the IBI362 6 mg cohort, with a reported increase in HbA1c from 7.6% at baseline to 8.4% at week 12 despite a 6.5% body weight reduction (Figs. 2D and S3B).

With respect to secondary efficacy endpoints, IBI362 treatment was associated with reduction in post-prandial glucose levels, increased post-MTT insulin excursion, C-peptide levels, and oral glucose insulin sensitivity, similar to those with dulaglutide treatment. More detailed assessment on glucose excursions using seven-point glucose measurements or continuous glucose monitoring would be helpful to provide further insights into the glycemic effects of IBI362 dual agonist. Nonetheless, the available data allay the concern of glucagon's stimulatory effect on hyperglycemia.

Unfortunately, there were no data on HOMA-beta or HOMA-IR to better assess changes in insulin resistance and insulin sensitivity.

***Response:** We thank the reviewer for the comments. IBI362 has been designed to balance the agonism of GLP-1 receptor and glucagon receptor to achieve dual benefits of glucose-lowering and weight loss. IBI362 showed promising glucose-lowering and weight loss efficacy in this study, while effect of outliers was also evident. Obviously, the efficacy of IBI362 in patients with type 2 diabetes needs to be validated in a larger population. The improvement in glycaemic control was evidenced by reductions in HbA_{1c} levels, postprandial glucose levels, as well as results from self-monitoring of blood glucose, assessed at baseline and week 12. The 7-point SMBG levels were markedly reduced in the IBI362 groups and dulaglutide group, compared with placebo group. We added the result of SMBG as Fig. S6, with description in the Methods and Results accordingly (Lines 166-167, 301-302).*

Further analysis on HOMA- β and HOMA-IR indicated numerical improvements in insulin sensitivity and resistance, while small sample size precluded robust conclusions and clinical interpretation. We added the results of HOMA- β and HOMA-IR as Fig. S4, with description in the Methods and Results accordingly (Lines 169-170, 294-296).

IBI362 treatment was associated with a modest reduction in total cholesterol, LDL-cholesterol and triglyceride levels similar to dulaglutide treatment (Fig. S6). ALT and AST liver enzymes generally decreased slightly from baseline to week 12 with IBI362 treatment, with one participant in the 3 mg cohort showing a transient increase in ALT level that peaked at week 4.

An unexpected finding was the significant suppression rather than augmentation of endogenous GLP-1 and oxyntomodulin levels in fasting and post-prandial states with all 3 doses of IBI362 treatment (Fig S4F and S4G). The authors posited “a feedback inhibition mechanism of gastrointestinal hormone pathways”. This observation clearly requires further investigation and confirmation in a larger and different patient population other than Chinese.

***Response:** We thank the reviewer for the comments. Despite the great advances in the development of GLP-1 receptor agonists and poly-agonists, little is known about the changes in incretins and gastrointestinal hormones in response to these exogenous agonists. Cotadutide, a daily-dose GLP-1 and glucagon receptor dual agonist, suppressed active and inactive forms of GLP-1, as well as glucagon, in both fasting and postprandial settings, in obese or overweight patients with type 2 diabetes². As IBI362 was designed as an oxyntomodulin analogue, we are interested in the changes of endogenous oxyntomodulin in patients treated with IBI362. At week 12, both fasting and postprandial GLP-1 and oxyntomodulin were markedly suppressed. While the changes in GLP-1 were generally consistent with those observed in patients with cotadutide, we agree with the reviewer that changes in oxyntomodulin warrant further mechanistic investigation in a wider population and in a long-term study. We made a revision in the Discussion (**Lines 379-387**).*

The rationale for pursuing the development of a dual GLP-1 and glucagon receptor co-agonist is to exploit the stimulatory effect of glucagon on energy expenditure, and which might lead to greater weight loss than what can be achieved through central appetite suppression with a GLP-1 receptor agonist alone. Glucagon infusion in humans and rodents increased oxygen consumption, which is presumably driven by its thermogenic effects on the liver and adipose tissue. This study did not investigate the effect of IBI362 treatment on energy expenditure using doubly labelled water, respirometry, oxygen consumption, or even with activity monitors, which represents a

missed opportunity. Glucagon may exert catabolic effect on amino acid and protein metabolism, and could potentially have undesirable effect on muscle and lean body mass. Hence body composition measurements with dual GLP-1 and glucagon receptor co-agonist are useful tools to assess lean muscle and body fat mass.

Response: We thank the reviewer for the great recommendation and agreed that energy expenditure and body composition measurements, as well as metabolite profiling are of great significance in the mechanistic investigation of IBI362. As the early phase trial, the main objective was the safety and tolerability. Some mechanistic investigations were planned in phase 2 studies of IBI362 in Chinese patients with overweight or obesity, and those with type 2 diabetes.

Significance

Overall this small, short-term phase 1b study with IBI362, a novel weekly GLP-1 and glucagon receptor dual agonist, demonstrated an acceptable safety and efficacy profile. IBI362 once weekly treatment also led to fairly robust improvements in glycemic control and body weight loss and these effects appear to be greater than those reported in a 54-week phase 2b randomized trial on cotadutide, a daily GLP-1 and glucagon receptor dual agonist under development (Nahra R et al. Diabetes Care 2021;44:1433-42). The more pronounced increase in heart rate expected in a dual agonist, which has also been reported by other dual GLP-1 and glucagon receptor agonists under development, remains a safety concern. The magnitude of glucose-lowering and weight loss benefits of IBI362 treatment reported in this manuscript is similar, and perhaps marginally better than those of such long-acting GLP-1 receptor agonists as liraglutide and semaglutide, both of which are currently approved for diabetes and obesity treatment. Current data warrant further evaluation of this novel dual agonist in a phase 2 trial, which will provide more clarity on its potential development as a treatment option for obesity and T2D.

Reviewer #2 (Remarks to the Author):

Summary: This paper describes a randomized, placebo-controlled, multiple ascending dose phase 1b study to evaluate the safety, tolerability, pharmacokinetics, and efficacy of a novel weekly-dose GLP-1 and glucagon receptor dual agonist (IBI362) for the treatment of type 2 diabetes and obesity. This study compares the performance of three different ascending dose regimens for IBI362 to placebo and an active reference treatment (dulaglutide). The study concluded that the safety profile for all three dose regimens of IBI362 to be acceptable, that the efficacy of IBI362 for glycemic control was comparable to the active reference dulaglutide, and that reductions in body weight were greater for IBI362 compared to the active reference dulaglutide. Overall, this paper provides a meaningful contribution to the field regarding treatments for diabetes and obesity. However, there are a few minor issues that should be clarified prior to acceptance for publication:

1. The inclusion criteria for this study restrict the sample to patients whose diabetes was inadequately controlled with diet and exercise alone or with stable metformin therapy. For those patients who were using metformin therapy prior to enrollment in the study, was metformin therapy stopped prior to enrollment in the study, or were the study treatments added to the existing metformin therapy? If metformin therapy was stopped prior to enrollment in the study, would the 3-week screening period be considered a sufficient “wash-out period” to ensure that the effects of the metformin therapy would not influence the results?

Response: We thank the reviewer for the comments. Patients were required to continue their pre-trial dose of metformin throughout the trial. We made the clear in the Methods (Lines 136-138).

2. In the “Randomization and masking” section, it is stated that the active reference dulaglutide group was open-label (line 115). However, it also states that “the study drugs and placebo were identically labeled and indistinguishable in appearance” (lines 117 – 118). These two statements seem to be contradictory. Please clarify whether it was known who was in the dulaglutide group, or if this group was masked in a similar manner as the IBI362 and placebo groups.

Response: We apologized for the inaccurate expression. The double-blindness applied only to IBI362 and placebo, whereas dulaglutide was open-label. We revised the

wording in the Methods (Lines 123-126).

3. On lines 163 – 164 in the “Statistical analysis” section, it states that “the sample size in this study was empirically determined to provide adequate safety, tolerability and pharmacokinetics data.” It is unclear what this statement means. Please clarify how the sample size was determined for this study.

***Response:** We thank the reviewer for the comments. As a phase 1 trial, the primary objective is to assess the safety and tolerability. The sample size for each dose group was determined empirically. A sample size of 8-12 subject per dose cohort would be sufficient to provide preliminary safety data and was most common for the drug class in early phase studies. In this study, 12 patients were assigned to IBI362 or placebo in an 8:4 ratio, and a total of six patients received open-label dulaglutide to provide a reference for safety and efficacy. The sample size was not powered for any efficacy endpoint. We made a revision in the Methods (Line 172-174).*

4. On line 165 in the “Statistical analysis” section, it states that the placebo and dulaglutide groups were pooled for analysis. However, this statement seems inconsistent with what is presented in the results; all tables and figures appear to present results separately for the placebo and dulaglutide groups. Please clarify whether these groups were pooled for analyses, and if so then for which analyses.

***Response:** We apologize for the inaccurate expression. In each cohort, four patients were assigned to placebo and two to dulaglutide. Thus, patients receiving placebo (n = 12) and dulaglutide (n = 6) were each pooled for analysis. We revised the wording in the Methods (Line 175).*

5. On line 201 in the Results section, it is stated that of the 42 patients in the safety and efficacy population, 37 patients completed the treatment. For patients who discontinued the study treatment prior to completing or discontinuing the study, were outcome measures collected for these patients following treatment discontinuation, and if so were data following treatment discontinuation included in statistical analyses (i.e., for intention-to-treat analyses)?

***Response:** We thank the reviewer for the comments. In this study, among five patients who discontinued the treatment, four quitted the study on study drug discontinuation, and one missed the week 12 dosing due to COVID-19 pandemic but completed the*

safety follow-up visit. The safety data of this patient was collected and included in the analysis. We made this clear in the Results (Lines 210-213).

6. For the tables that present treatment emergent adverse events (Table 2 and Supplementary Table 2), do the numbers in the table correspond to the number of events, or the number of patients who experienced at least one event?

Response: *Safety data in this study were presented as number of patients (percentage). We made this clear in the footnotes (Fig. 2 and S3) of relevant tables and apologized for the neglect.*

7. For Supplementary Table 1: Why is the terminal elimination half-life outcome ($t_{1/2}$) not included in this table?

Response: *In the single dose range of 0.03-5 mg, IBI362 showed a linear pharmacokinetic profile, and the half-life was not affected by the dose. Thus, we combined the half-life result for the three dose groups and presented it in the main text (Lines 270-271).*

Reviewer #3 (Remarks to the Author):

Jiang report on a phase 1 b clinical trial with a novel dual GLP-1/glucagon co-agonist in Chinese patients with type 2 diabetes. This compound (IBI362/LY3305677) is used with once-weekly injections. Compounds addressing several pancreatic or intestinal hormone receptors are an important area of research to improve glycaemic control and body weight in subjects with type 2 diabetes and obesity. The present study reports substantial improvements in glycated haemoglobin and body weight, indicating some therapeutic potential for the compound studied. However, results with placebo treatment indicate significant changes vs. baseline. This complicates the interpretation. The overall number of patients studied was low (6-8 per group with active drugs). This explains some of the problems mentioned in detail below.

Major points:

(1) The main outcomes should be reported as placebo-corrected effects. If the statistical analysis plan requires to report the effects as they occurred, the placebo-corrected values should be added as a meaningful post-hoc analysis. The fact that there were no clear dose-response relationships should be explained.

Response: We thank the reviewer for the comments. Estimation of treatment difference versus placebo of each IBI362 group and dulaglutide group in main efficacy endpoints (HbA_{1c}, fasting plasma glucose and body weight) was defined in the statistical analysis plan. We added the results of changes from baseline in HbA_{1c}, fasting plasma glucose and body weight in each treatment group and treatment differences versus placebo as a supplementary table (Table S2), with description of relevant result (Lines 279-281, 292-294, 312-314). The methods for the analysis was clearly stated in the Methods (Lines 196-199).

No clear dose-response relationship was observed in terms of the aforementioned efficacy endpoints. Firstly, the randomization was done within each cohort. With small number of patients, baseline characteristics were not strictly balanced. Secondly, the small sample size, together with some outliers, added substantial variations in each group, which precluded robust statistical analysis and interpretation. Third, the 12-week treatment period was short for evaluation of glucose-lowering and weight loss efficacy, both of which did not reach plateau at week 12. Thus, a phase 2 study with larger sample size and long duration has been

launched to evaluate the efficacy of IBI362. We added a discussion in the main text (Lines 372-376).

(2) The rationale behind stimulating glucagon as well as GLP-1 receptors is additional weight loss (increases in energy expenditure due to glucagon agonism). It is unclear why weight reduction was treated as exploratory endpoint.

Response: We thank the reviewer for the comment. Two phase 1b clinical trials were done to assess the safety and efficacy of IBI362 in participants with overweight or obesity and patients with type 2 diabetes, respectively. Chinese patients with type 2 diabetes have relatively lower body weight and BMI of compared with the western population. With eligible BMI of 20-35 kg/m², this study was primarily designed to evaluate the safety and glucose-lowering efficacy of IBI362 in Chinese patients with type 2 diabetes.

(3) Line 165: The statement that “patients receiving placebo or dulaglutide were pooled for analysis” is unclear and may be misleading. Placebo and dulaglutide should not be lead to similar effects.

Response: We apologize for the inaccurate expression. In each cohort, four patients were assigned to placebo and two to dulaglutide. Thus, patients receiving placebo (n = 12) and dulaglutide (n = 6) were each pooled for analysis. We revised the wording in the Methods (Line 175).

(4) The ECG abnormalities did newly occur during treatment with IBI362/LY3305677?

Response: All cardiac disorders reported in patients receiving IBI362 were revealed by electrocardiogram. Except for the non-serious adverse event of arrhythmia and serious adverse event of myocardial ischemia reported in one patient receiving IBI362 in the 3.0 mg cohort, all other cardiac disorders were mild in severity. We scrutinized the medical history of the patients who reported cardiac disorders during the study. Both patients reporting serious cardiac adverse events had hypertension and one had abnormal T-wave at screening. One patient reporting atrioventricular block first degree in the IBI362 6.0 mg group had hypertension and Sinus bradycardia as revealed by electrocardiogram at screening.

(5) The use of modelling to estimate insulin sensitivity probably is misleading when

using an agent that likely decelerates gastric emptying: This should result in lower glucose increments with lower insulin response, a combination which will be interpreted as an increased insulin sensitivity. See also discussion (lines 339-341).

*Response: We thank the reviewer for the comment. We deleted the analysis regarding oral glucose insulin sensitivity and presented HOMA- β and HOMA-IR instead (Fig. S4 and **Lines 294-296**).*

Minor points:

(a) Line 40 ff. and elsewhere: Since nausea and vomiting are most likely caused by a direct interaction of the agent with the brain stem and does not involve functional changes in the GI tract, the term “gastrointestinal disorders” should be replaced by “gastrointestinal symptoms (nausea, vomiting and diarrhoea)”. The abstract should present numbers regarding weight loss (line 46). The low frequency of vomiting is remarkable.

Response: We thank the reviewer for these comments. We agreed with the reviewer and used “gastrointestinal symptoms” or “gastrointestinal adverse events” when applicable. However, in describing incidence of adverse events, we used the MedDRA SOC “gastrointestinal disorders” instead.

*We added numbers for weight loss efficacy in the abstract (**Lines 51-53**).*

*We also revised the discussion of gastrointestinal adverse events. (**Lines 331-336**). The relative frequency of individual GIAE was inconclusive with small sample size.*

(b) Line 68: typo: Addictive should be additive

*Response: We apologized for the neglect and made revision accordingly (**Line 74**).*

(c) The introduction mentions a previous multiple ascending dose study with this agent in Chinese adults. The rationale for performing a second study should be mentioned along these lines. The identification of optimal dosing regimens is usually the main objective in phase 2.

Response: We thank the reviewer for the comment. As mentioned above, two phase 1b studies explored the safety and efficacy of IBI362 in Chinese patients with overweight or obesity and patients with type 2 diabetes, respectively. These two studies adopted similar study design and same dose regimens of IBI362. We revised this part to

provide a clearer background (**Lines 98-101**).

(d) What is the purpose of measuring amidated GLP-1 (line 137)? This can still be active (7-36 amide) or inactive (9-36 amide), so why not measure total or intact GLP-1?

Response: *We thank the reviewer for the comment. We used Mercordia Total GLP-1 NL-ELISA kit (Cat No. 10-1279-01) to measure GLP-1 in this study. The kit was designed to detect GLP-1 (9-36) amide, with 93% crossreaction to GLP-1 (7-36) amide and 88% crossreaction to GLP-1 (1-36) amide. We revised the description to make this clear (**Lines 144-145**).*

(e) Why did dulaglutide not change weight?

Response: *We referred to a phase 2 trial of tirzepatide⁸ and a phase 3 trial of semaglutide (SUSTAIN 7)⁹, both of which used dulaglutide 1.5 mg as a reference. The 12-week weight loss in patients with dulaglutide 1.5 mg was mild in these studies and similar to that observed in this study.*

(f) The detailed report of results (lines 331-338) should be moved to the results section

Response: *We agreed with the reviewer and made revision (**Lines 284-288**).*

Michael A. Nauck

Reference:

1. Ji L, *et al.* IBI362 (LY3305677), a weekly-dose GLP-1 and glucagon receptor dual agonist, in Chinese adults with overweight or obesity: A randomised, placebo-controlled, multiple ascending dose phase 1b study. *EClinicalMedicine* **39**, 101088 (2021).
2. Ambery P, *et al.* MEDI0382, a GLP-1 and glucagon receptor dual agonist, in obese or overweight patients with type 2 diabetes: a randomised, controlled, double-blind, ascending dose and phase 2a study. *The Lancet* **391**, 2607-2618 (2018).
3. Parker VER, *et al.* Efficacy, Safety, and Mechanistic Insights of Cotadutide, a Dual Receptor Glucagon-Like Peptide-1 and Glucagon Agonist. *J Clin Endocrinol Metab* **105**, (2020).
4. Nahra R, *et al.* Effects of Cotadutide on Metabolic and Hepatic Parameters in Adults With Overweight or Obesity and Type 2 Diabetes: A 54-Week Randomized Phase 2b Study. *Diabetes Care*, (2021).
5. Tillner J, *et al.* A novel dual glucagon-like peptide and glucagon receptor agonist SAR425899: Results of randomized, placebo-controlled first-in-human and first-in-patient trials. *Diabetes Obes Metab* **21**, 120-128 (2019).
6. Alba M, Yee J, Frustaci ME, Samtani MN, Fleck P. Efficacy and safety of glucagon-like peptide-1/glucagon receptor co-agonist JNJ-64565111 in individuals with obesity without type 2 diabetes mellitus: A randomized dose-ranging study. *Clin Obes* **11**, e12432 (2021).
7. Ambery PD, *et al.* MEDI0382, a GLP-1/glucagon receptor dual agonist, meets safety and tolerability endpoints in a single-dose, healthy-subject, randomized, Phase 1 study. *Br J Clin Pharmacol* **84**, 2325-2335 (2018).
8. Frias JP, *et al.* Efficacy and safety of LY3298176, a novel dual GIP and GLP-1 receptor agonist, in patients with type 2 diabetes: a randomised, placebo-controlled and active comparator-controlled phase 2 trial. *The Lancet* **392**, 2180-2193 (2018).
9. Pratley RE, *et al.* Semaglutide versus dulaglutide once weekly in patients with type 2 diabetes (SUSTAIN 7): a randomised, open-label, phase 3b trial. *The Lancet Diabetes & Endocrinology* **6**, 275-286 (2018).

REVIEWERS' COMMENTS

Reviewer #1 (Remarks to the Author):

The authors have satisfactorily addressed each of the concerns raised by the reviewers' in the revised manuscript.

Reviewer #2 (Remarks to the Author):

Thank you for revising the manuscript. The manuscript is much clearer now. However, a couple minor issues still need to be clarified:

1. It is still unclear in the "Statistical analysis" section of the manuscript what is meant by "the sample size in this study was empirically determined..." Can you please be more specific by what you mean by "empirically determined"? Does this mean that the sample size was determined based on sample sizes commonly used in previous early phase studies of medications in the same drug class? If so, please state that in the manuscript.
2. In the "Statistical analysis" section, it is still unclear what is meant by "patients receiving placebo and dulaglutide were each pooled for analysis." This sounds to me as though the placebo and dulaglutide groups were pooled together for analysis (i.e., as if outcomes were summarized for a single combined placebo and dulaglutide group of size $n=18$), but based on the results it looks as though the placebo ($n=12$) and dulaglutide ($n=6$) groups were each analyzed separately. Is it meant that the placebo group was pooled across the three sample cohorts (and the same is true for the dulaglutide group)? If so, then I would suggest revising the sentence as similar to the following: "Patients receiving placebo ($n=12$) and dulaglutide ($n=6$) were each separately pooled across the three sample cohorts for analysis."

Reviewer #3 (Remarks to the Author):

Jiang et al. have responded appropriately to all points raised by the reviewers, which led to an extensive revision, which certainly has improved the quality of the manuscript considerably.

Michael Nauck

REVIEWER COMMENTS

Reviewer #2 (Remarks to the Author):

Thank you for revising the manuscript. The manuscript is much clearer now.

However, a couple minor issues still need to be clarified:

1. It is still unclear in the “Statistical analysis” section of the manuscript what is meant by “the sample size in this study was empirically determined...” Can you please be more specific by what you mean by “empirically determined”? Does this mean that the sample size was determined based on sample sizes commonly used in previous early phase studies of medications in the same drug class? If so, please state that in the manuscript.

Response: We thank the reviewer for the comments. We revised the sentence in the Methods (Lines 384-385).

2. In the “Statistical analysis” section, it is still unclear what is meant by “patients receiving placebo and dulaglutide were each pooled for analysis.” This sounds to me as though the placebo and dulaglutide groups were pooled together for analysis (i.e., as if outcomes were summarized for a single combined placebo and dulaglutide group of size $n=18$), but based on the results it looks as though the placebo ($n=12$) and dulaglutide ($n=6$) groups were each analyzed separately. Is it meant that the placebo group was pooled across the three sample cohorts (and the same is true for the dulaglutide group)? If so, then I would suggest revising the sentence as similar to the following: “Patients receiving placebo ($n=12$) and dulaglutide ($n=6$) were each separately pooled across the three sample cohorts for analysis.”

Response: We thank the reviewer for the comments. The proposed description is more clear. We made the revision accordingly (Lines 388-389).